# Air Quality Standards and Extreme Ozone Events in the São Paulo Megacity

**Júlio Barboza Chiquetto [1,\*]**, **Maria Elisa Siqueira Silva [2]**, **William Cabral-Miranda [2]**,
**Flávia Noronha Dutra Ribeiro [3]**, **Sergio Alejandro Ibarra-Espinosa [4,5]** and **Rita Yuri Ynoue [5]**

[1] Institute of Advanced Studies, University of São Paulo, Sao Paulo 05508-050, Brazil
[2] Department of Geography; University of São Paulo, Sao Paulo 05508-000, Brazil
[3] Department of Environmental Management; University of São Paulo, Sao Paulo 03828-000, Brazil
[4] Key Laboratory of Wetland Ecology and Environment, Chinese Academy of Sciences,
   Changchun 130102, China
[5] Department of Atmospheric Sciences, University of São Paulo, Sao Paulo 03178-200, Brazil
**\*** Correspondence: julio22@usp.br; Tel.: +55-11-99452-4010

**Abstract:** Ozone events in South America might be triggered by increasing air temperatures and dry conditions, leading to vulnerable population exposure. The current air quality standards and attention levels in São Paulo state, Brazil, are 40% higher and 25% higher, respectively, than the limits recommended by the World Health Organization (WHO). We simulated an extreme ozone event in the São Paulo megacity using the Weather Research and Forecast/Chemistry model during an extreme event characterized by positive anomalies of air temperature and solar radiation. Results were evaluated using the different air quality limits from São Paulo state and the WHO, also with socioeconomic vulnerability data from the Brazilian census and cost analysis for the public health system from the extreme episode. More than 3 million people in vulnerability conditions, such as low income and families with an above-average percentage of children, live in areas where ozone concentrations exceeded the attention levels of the WHO during the episode, which is ignored by the lenient SP state environmental laws. WHO air quality guidelines must be adopted urgently in developing nations in order to provide a more accurate basis for cost analysis and population exposure, particularly the for vulnerable population groups.

**Keywords:** air pollution; air quality modeling; ozone; urban environment; São Paulo; air pollution exposure; extreme events; vulnerability; environmental governance; cost analysis

## 1. Introduction

Air pollution represents health risks and an overall decrease in quality of life, particularly in densely populated megacities. Pollutants impact human health, but the source-receptor pathway is intermediated by atmospheric conditions and population exposure. The World Health Organization (WHO) has been playing a vital role in defining air quality standards for criteria air pollutants, based on several international epidemiological studies, defined as concentration thresholds above which adverse health effects are expected to occur to a significant parcel of the population [1,2]. Currently, the ozone ($O_3$) limits in the state of São Paulo (SP) are higher than the WHO guidelines (Table 1).

In 2017, the state ozone standard was exceeded on 28 days in the metropolitan area of São Paulo (MASP), in 22 of 23 monitoring sites. Under these conditions, high population exposure to air pollution is expected, which is not correctly assessed by the lenient environmental laws in the MASP.

Vehicles are the greatest source of air pollution in this region [3]. However, pollutants can be transported and affect areas tens of kilometers downwind. This is important for ozone, which is

formed by photochemical reactions 1 or 2 hours after the emission of its precursors (NOx and VOCs). These reactions are triggered by sunlight and can be potentialized by higher air temperatures, above 20 °C [4]. In the MASP, atmospheric conditions favorable for ozone concentrations have been observed during many periods in the previous years [5,6]. According to a broad scientific literature, the frequency and intensity of extreme climatic events are likely to increase in the following decades, so, high ozone episodes are also likely to increase in this region [7].

As other megacities in the developing world, the MASP is a dense urban conglomeration of 39 cities, 23 million inhabitants, concentrating nearly 10% of the national GDP and marking socioeconomic contrasts. The spatial variation of life expectancy in the municipality of São Paulo alone ranges from 58 to 81 years of age [8]. Several studies worldwide investigated air pollution exposure according to socioeconomic conditions, showing that the lower-income population, elderly and children are more vulnerable to the effects of air pollution [9–12], similarly to São Paulo [13–15]. Recent studies on vulnerability to high temperature and air pollution demonstrated how the intraurban spatial variation of environmental conditions are associated to exposure and health effects, but additional studies are needed for a better assessment of these relationships [16]. During high ozone episodes in São Paulo, most of the population experiences very warm and sunny conditions. The lower income population, besides living in poor housing conditions with worse wall insulation, tends to keep their doors and windows open to promote better air circulation and improve thermal comfort. However, this increases outdoor to indoor air exchanges, and so, population exposure to ambient pollution increase inside their homes [9].

According to previous studies, increasing ozone concentrations lead to an increased risk of respiratory problems [2]: for every 10 $\mu gm^{-3}$ increase in ozone, the relative risk of overall mortality increases by 0.1% [17]. In areas such as the MASP, the population density reaches more than 50,000 persons/km$^2$ in the most crowded districts (Figure S1). Therefore, defining limits which are higher than the guidelines of the WHO (Table 1) poses a serious risk to millions of people, impairs proper environmental and health planning and indicates a lack of responsibility by the decision makers. High ozone concentrations potentially worsen asthma conditions and lead to ER visits and hospitalization [1,17,18].

**Table 1.** Air quality standards and attention levels used in the study: São Paulo (SP) Standard, World Health Organization (WHO) Standard, SP Attention Level and WHO Attention Level.

| Standard (8-h Ozone Average) | WHO | SP | Difference |
|---|---|---|---|
| Air Quality Standard | 100 $\mu gm^{-3}$ | 140 $\mu gm^{-3}$ | +40% |
| Attention Level | 160 $\mu gm^{-3}$ | 200 $\mu gm^{-3}$ | +25% |

In the US, ozone events have been estimated to lead to billions of dollars of economic loss yearly [18]. In Brazil, direct hospital asthma costs alone have been estimated to be more than US$500.00 per patient yearly, not to mention the more systemic economic impacts of medication needs and absenteeism [19]. A study comparing health and economic impacts using actual ozone averages and those recommended by the WHO shows that, by reaching the WHO levels, 50 yearly hospitalizations and 152 premature deaths would be avoided in São Paulo, leading to an overall economic gain of US$24.9 million [20]. Most studies on health costs use average exposure conditions. However, extreme air pollution events certainly correspond to a larger percentage of yearly average exposure. For example, events of bushfire smoke have been investigated to produce health effects on more than 30,500 people in the municipality of Albury, Australia, over the course of only 30 days [21]. The Fuzzy Technique for Order Preference by Similarity to Ideal Situation (TOPSIS) has been used to evaluate vulnerability to air pollution and climate change in South Korea, but not using air quality modelling [22]. Other integrative works using air quality modelling have been performed for other regions, such as the Economic Valuation of Air Pollution (EVA) model for Denmark and Europe [23]. However, it focused on a continental scale, and few studies have been performed for the metropolitan scale. Policymakers must be aware of

the real risks faced by the population in order to design and implement health and environmental management strategies which are both efficient and cost-effective.

The objective of this work is to show how outdated air quality standards lead to incorrect assessments of the impacts of an extreme ozone event. We show the number of exceedances of the air quality standards and attention levels, in the area size, the difference in the number of people affected, including those in vulnerable situations and cost analysis for the public health system. Results are shown using the current air quality standards and attention levels from the SP state, compared to the guidelines recommended by the WHO. We discuss the consequences of the lenient air quality limits of the SP State for environmental and health management and planning.

## 2. Materials and Methods

### 2.1. WRF/Chem Simulation

The summer of 2014 was characterized by high irradiance, air pressure and air temperature in the southeast region of Brazil, with positive temperature anomalies of 4 °C and the total accumulated precipitation in summer of only 32% of the climatological total for the region, which is usually the wettest season [5]. The strengthening of the Subtropical Atlantic High-Pressure System inhibited the passing of cold fronts and the development of precipitation events in the region, which favored air pollutants concentration, particularly $O_3$ [6]. As a result, there were 43 days with exceedances of the state $O_3$ standards in 2014. We chose the period from 28/01/2014 00Z to 02/02/2014 00Z for simulation, when 24 exceedances of the ozone standards were observed in the MASP [24]. Due to the continental heating, this period presented intense regional temperature contrasts between the ocean and the continent, which favored the local circulation system of sea-land breeze, one of the most typical local wind systems in São Paulo [25,26].

We performed a simulation of the high ozone episode using the atmospheric model WRF/Chem version 3.2.1 [27] with the emission scheme of Andrade et al. [28] which uses a bottom-up approach similar to other modelling tools [29]. We chose this version of the model due to the specific VOCs treatment available in this version, which considers ethanol as an explicit species. This is appropriate for this study area due to the massive use of ethanol as vehicle fuel in Brazil, and for representing ozone atmospheric chemistry. Vehicular emission estimates were based on emission factors published by the environmental agency of the state of São Paulo, from tunnel observation experiments [28]. Pollutant emission is proportional to total road length inside each 1-km$^2$ grid cell. More detailed information about the model can be found in Figure S3, Table S1 and Appendix A. We described the model and its validation in another publication [6].

### 2.2. Air Quality Standards

Results were evaluated considering the exceedances of the ozone air quality standards and attention levels from the state of São Paulo (SP) and the World Health Organization (WHO). To check if the chosen threshold is exceeded, an eight-hour average is calculated, considering the previous seven hours. This calculation was performed for all hours of the simulation period, according to Equations (1)–(4):

$$\text{AQSE SP } = \frac{T_H + T_{H-1} + T_{H-2} + \ldots + T_{H-7}}{8} \geq 140 \ \mu\text{g·m}^{-3} \tag{1}$$

$$\text{ATTE SP } = \frac{T_H + T_{H-1} + T_{H-2} + \ldots + T_{H-7}}{8} \geq 200 \ \mu\text{g·m}^{-3} \tag{2}$$

$$\text{AQSE WHO } = \frac{T_H + T_{H-1} + T_{H-2} + \ldots + T_{H-7}}{8} \geq 100 \ \mu\text{g·m}^{-3} \tag{3}$$

$$\text{ATTE WHO } = \frac{T_H + T_{H-1} + T_{H-2} + \ldots + T_{H-7}}{8} \geq 160 \ \mu\text{g·m}^{-3} \tag{4}$$

where:

AQSE = air quality standard exceedance
ATTE = attention level exceedance
$T_H$ = concentration at the analyzed hour
$T_{H-n}$ = concentration n hours earlier than the analyzed hour
SP = State of São Paulo
WHO = World Health Organization

These four standards were used to compare the difference between: (1) the number of exceedances of each limit; (2) the size of the area of exceedance of each limit; (3) the number of persons affected under each limit and (4) the estimated potential costs for the public health system. In order to produce the results as maps, we separated the ozone concentration values above each threshold from the rest of the time series and averaged these values, resulting in one averaged map for each limit, indicating which areas in the domain exceeded that limit during the episode. To calculate the size of these areas of exceedance, we converted the model output Network Common Data Form (NETCDF) files into shapefiles.

### 2.3. Socioeconomic and Health Vulnerability

We used socioeconomic data from the last Brazilian census available (2010) in order to assess potential population exposure to the high ozone concentrations during the extreme episode. Previous exposure studies with different types of modelling in São Paulo used grid cells of 750 m [15] or 2 km [13], therefore, we used the weighting areas from the census, which are larger aggregations of the census tracts by population, for a better match with the WRF/Chem spatial resolution grid of 1 km$^2$. The maps with the areas of exceedance of each limit were joined with the MASP weighting areas in order to obtain the estimated number of residents inside the area of exceedance of each limit. To estimate the number of people in vulnerability to air pollution, we chose five variables: percentage of elderly people, percentage of children, percentage of low-income families, percentage of people in houses with no external coating and percentage of people diagnosed with asthma (Table 2).

**Table 2.** Vulnerability criteria used in the study.

| Vulnerability Criterion | Explanation |
|---|---|
| Low income | Monthly family income equal to or lower than US\$500 [30]. |
| Elderly | Persons with 60 years of age or more. |
| Children | Persons with 5 years of age or less. |
| No external coating | Persons living in houses made of bricks without external isolation. |
| Asthmatics | Approximately 5% of the MASP population. [1] |

[1] according to data from the National Health Survey (from the 2010 Census).

To estimate the number of low-income people living in the areas of exceedance, we used the new international poverty line for upper middle countries of the World Bank [30], which is US\$5.5 per person/day. We multiplied this value by three (according to the Brazilian average of three persons per family) and then multiplied by 30 (number of days in a month), reaching an average of US\$500 (equivalent to approximately two minimum wages in Brazil in 2019, R\$1908.00). The number of elderly people (with 60 years of age or more) and children (with 5 years of age or less) were calculated from the percentage of these groups among the total population in each weighting area. The number of people living in houses with no external coating was calculated based on the percentage of houses without external coating. The number of asthmatics was calculated using data from the national health survey available online [31]. We chose the variable "percentage of the total population diagnosed with asthma" in the MASP, which yielded 5% of its population. In lack of any more in-depth data, we assumed this condition to be evenly distributed among the MASP weighting areas. We then compared the areas with ozone exceedances of the different limits with the areas containing people in

these conditions (Figure 2). Low income and population density maps for the MASP can be found in Figures S1 and S2.

## 2.4. Simplified Cost Analysis

Ozone has long been associated with hospital admissions from asthma [32], although studies in Brazil are scarcer than those found in Europe or the US. To perform a simplified calculation of the potential costs for the public health system from asthma complications during the ozone episode, we used data from a previous study, in which the authors calculated the average cost of each emergency room visit and hospitalization event per patient due to effects of asthma in São Paulo [33]. We considered this cost as a public health system cost because the cost analysis was performed here for low-income populations, who cannot afford private healthcare. Moreover, from a governance point of view, it is the responsibility of Brazilian authorities to provide free universal healthcare. Considering that there was a reasonably high chance that each asthmatic person would visit ER at least once during the five days of the high ozone episode, we used the total estimated number of asthmatics in the areas with ozone standard exceedances and multiplied by the values for each ER visit in Santos et al. [33]. We considered the age-sensitive groups (elderly and children) to be as vulnerable as the asthmatics and also included them in this cost analysis.

For estimating the cost in the area of exceedance of the attention levels, we performed the same analysis, but adding the hospitalization event costs to the ER visits costs, considering that attention levels have been associated to more severe health effects [2,17]. We updated the values in Santos et al. comparing the reimbursement values from the period of that study with current values from the national health system available online (there was an increase in 5% in the values for procedures of chronic lower respiratory conditions) [34], along with 2019 exchange rates from R$ to US$, resulting in the cost estimates in Table 3 We used data from controlled asthma patients only.

**Table 3.** Simplified cost estimate values from health events derived from pollution events analyzed in this study.

| Ozone Pollution Event | Health Event/Asthmatic or Age-Sensitive Person | Average Value for Each Health Event [1] |
|---|---|---|
| Exceedance of ozone standard (SP state and WHO) | ER visit | US$1.74 |
| Exceedance of ATT (SP state and WHO) | ER visit + hospitalization | US$10.03 |

[1] Values from Santos et al. [33], updated to current costs and using a 2019 exchange rate of US$1 = R$3.9.

## 3. Results

We first analyzed the exceedances of the different ozone limits considering the total population.

### 3.1. Exceedances of Air Quality and Attention Levels in the MASP According to SP State and WHO Limits

Our simulation of the extreme ozone event resulted in 37 exceedances of the SP state ozone standard and of 55 exceedances of the WHO ozone standard, a 50% increase. For the attention levels, there was a greater difference, from five to 26, respectively (more than a fivefold increase). The difference in size indicates an area 30% larger for the exceedance of the WHO ozone standard (Figure 1a), and fourteen times larger when considering the WHO attention levels (Figure 1b).

Most areas of exceedance are located to the west and north of the MASP due to the transport of pollutants caused by the local circulation of the sea breeze coming from the southeast, during the afternoon [6,35]. We analyzed wind direction data in the available stations from the SP State Environmental Company during the month of January in 2014.The sea breeze influenced the MASP on 22 days of the month. The sea breeze is one of the most common local wind systems over the MASP [26]. As ozone is a secondary pollutant, formed one or two hours after the emission of its

precursors, it undergoes transport and high concentrations are commonly found in regions downwind, far from the emissions.

The SP ozone standard is exceeded over a large area in the central and west portions of the domain (yellow in Figure 1a). When the WHO ozone standards are considered, areas to the east of the MASP (red in Figure 1a) are also under violation of the air quality limits. However, by the SP state current legislation (8-h average ≥ 140 μgm$^{-3}$), these areas would not be considered priority areas during this episode, despite the fact that ozone concentrations above the WHO thresholds are associated to respiratory and cardiovascular diseases and overall mortality [17,18].

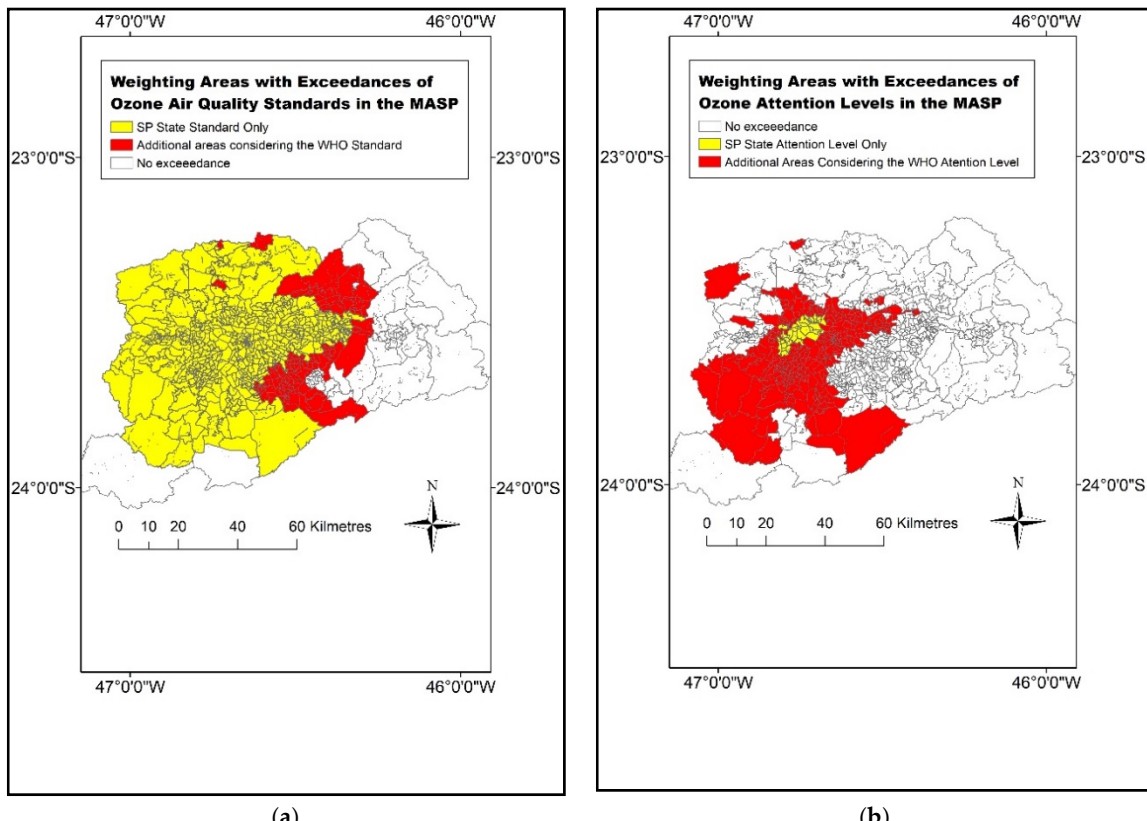

**Figure 1.** Weighting areas with exceedance of the (**a**) ozone standards from the São Paulo (SP) state (yellow: 140 μgm$^{-3}$) and the added areas when considering the World Health Organization (WHO) standard (red: 100 μgm$^{-3}$); (**b**) weighting areas with exceedance of the attention levels from the SP state (yellow: 200 μgm$^{-3}$) and the areas added when considering the WHO attention level (red: 160 μgm$^{-3}$), during the high ozone episode simulated (eight hour averages).

Concerning attention levels (Figure 1b), the differences are much greater. Only a small area in the center-west of downtown MASP experience exceedances of the SP state ozone standard (yellow in Figure 1b), but when the WHO attention level was considered, the exceedance occurred in a much greater area, fourteen times the size (Table 4), including many suburbs in the outskirts of the MASP (red in Figure 1b).

**Table 4.** Number of exceedances and area size of each standard during the high ozone episode.

| 8-h Ozone Average | 140 μgm$^{-3}$ | 100 μgm$^{-3}$ | 200 μgm$^{-3}$ | 160 μgm$^{-3}$ |
|---|---|---|---|---|
| **Standard** | **AQS SP** | **AQS WHO** | **ATT SP** | **ATT WHO** |
| Number of Exceedances (domain) | 37 | 55 | 5 | 26 |
| Size of Area (km$^2$) | 2931 | 4138 | 116 | 1656 |

AQS = air quality standard, ATT = attention levels, SP = State of São Paulo, WHO = World Health Organization.

### 3.2. Ozone Exceedances and Socioeconomic Vulnerability

In order to investigate ozone exposure in greater depth, we compared Figure 1a,b with socioeconomic vulnerability data (as explained in Section 2.3), resulting in Figure 2a,b.

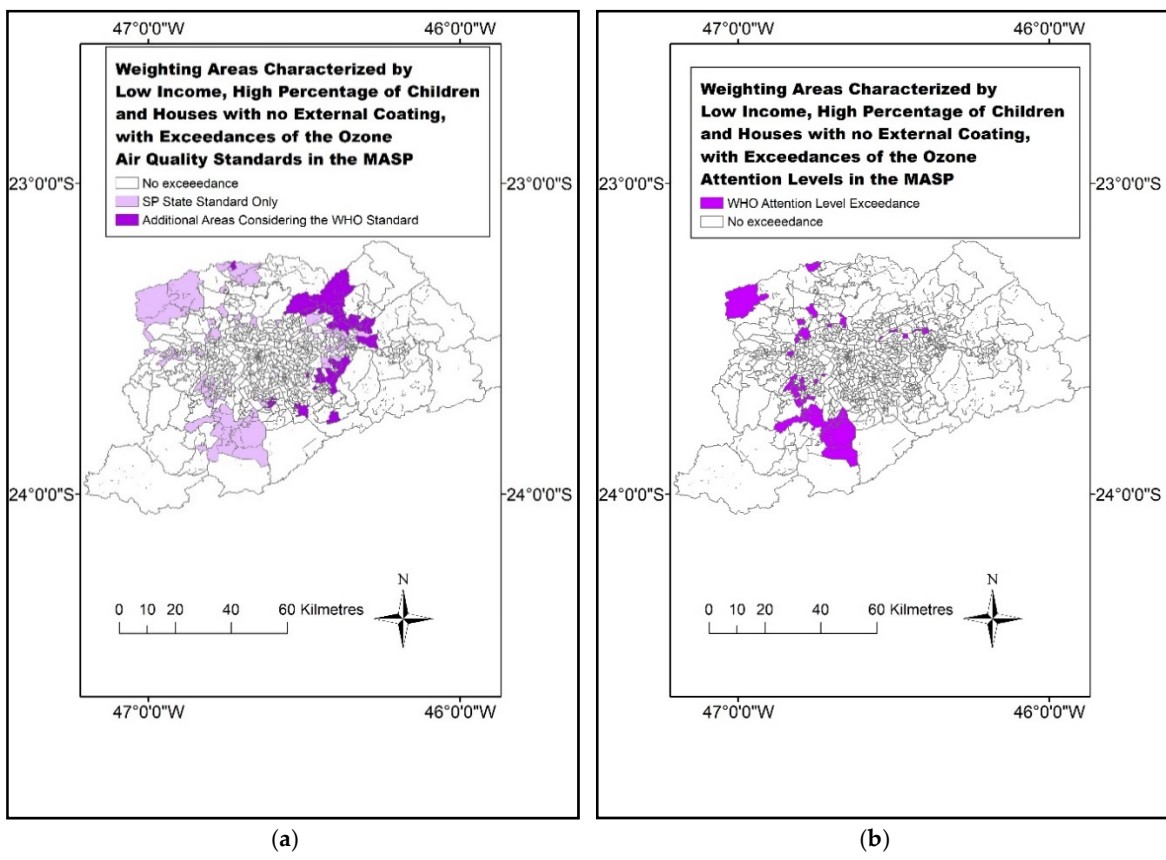

| (**a**) | (**b**) |

**Figure 2.** Weighting areas with low income population (monthly family income lower than US$500.00), higher than average percentage of children and houses with no external coating, with exceedance of the (**a**) ozone standards from the SP state (light purple: 140 μgm$^{-3}$) and the added areas when considering the WHO standard (dark purple: 100 μgm$^{-3}$); (**b**) weighting areas with exceedance of the attention levels from the WHO (dark purple: 160 μgm$^{-3}$), during the high ozone episode simulated (eight hour averages). No vulnerable areas exceeded the SP attention levels (200 μgm$^{-3}$).

The SP ozone standard is exceeded in many areas with vulnerable population outside the center of the MASP, in all directions, but mainly to the north and west (light purple in Figure 2a), patterns also driven by the sea breeze circulation. However, when using the stricter WHO limits, many vulnerable areas to the east were also included (dark purple in Figure 2a). The current attention level from the SP state was not exceeded in any of the vulnerable areas. However, the stricter attention level from the WHO was exceeded in many vulnerable areas in the MASP, mainly to the west, which were overlooked considering only the outdated SP Attention levels.

### 3.3. Number of People Affected

The number of potentially exposed people is shown in Table 5. We considered the vulnerable population groups as explained in Table 2; two additional groups were considered as extremely vulnerable, where one or more vulnerability criteria stacked and applied to the same population (in this case, low-income asthmatics with no external coating and low-income children and elderly).

The number of vulnerable people affected is considerable, surpassing one million asthmatics and people in houses with no external coating in areas with exceedance of the WHO ozone standard—an increase of 25% and 13%, respectively, compared to the SP standard. The absolute number of elderly and children affected was substantially smaller than the other groups but must still receive careful attention from the authorities. The number of low-income people affected by violations of the ozone standards was quite high: from 3.7 million (SP) to 5.3 million (WHO).

**Table 5.** Estimated population exposed and persons in vulnerability and extreme vulnerability, in the areas of exceedance of the four standards used.

| 8-h Ozone Average | 140 $\mu$gm$^{-3}$ | 100 $\mu$gm$^{-3}$ | 200 $\mu$gm$^{-3}$ | 160 $\mu$gm$^{-3}$ |
|---|---|---|---|---|
| Standard | AQS SP | AQS WHO | ATT SP | ATT WHO |
| **Total Population** | **16,187,395** | **20,233,937** | **1,045,076** | **9,726,100** |
| **Vulnerable Population** | | | | |
| Low income | 3,727,234 | 5,305,680 | 0 | 2,019,042 |
| Elderly + children | 93,169 | 107,587 | 6077 | 56,239 |
| No external coating | 1,435,927 | 1,673,286 | 48,475 | 776,048 |
| Asthmatics | 809,369 | 1,011,696 | 52,253 | 486,305 |
| **Extremely Vulnerable Population** | | | | |
| Low-income asthmatics with no external coating | 26,332 | 38,799 | 0 | 14,088 |
| Low-income elderly + children | 20,807 | 29,641 | 0 | 11,190 |

AQS = air quality standard, ATT = attention levels, SP = State of São Paulo, WHO = World Health Organization.

The number of people affected by the exceedance of the WHO attention levels was nearly ten times higher compared to the SP attention levels for the age-sensitive groups and asthmatics, and sixteen times higher for people in houses with no external coating. Concerning low income, there were more than 2 million people in areas with exceedance of the WHO attention levels, a risk which is completely ignored by the SP ozone standard (Figure 2b). The current SP state attention levels clearly underestimate the number of vulnerable people in areas susceptible to high ozone levels.

The number of people in extreme vulnerability affected was proportionally smaller. However, expressive differences still exist comparing SP and WHO levels. Low income elderly and children affected increased from 20,000 to nearly 30,000, a 50% increase considering the WHO ozone standard, not to mention the SP attention levels, which did not capture any risk for these population groups. Since they tend to spend most of their time at home, pollutant concentrations near their living spaces represent an even more accurate measure of their daily exposure compared to adults. It is important to consider these groups because they are much more likely to result in negative health outcomes.

### 3.4. Cost Analysis

We chose to only calculate the costs from those groups directly vulnerable to respiratory diseases due to physical/health reasons (asthmatics and age-sensitive groups). Substantial economic costs due to ER visits and hospitalizations from these groups are expected in extreme ozone events. Considering SP ozone standards, these add up to more than US$1.5 million in ER visits but reached nearly US$2 million if the WHO ozone standard was considered, nearly a 25% increase in the cost estimate (Table 6). For the cost estimate from the attention levels, the increase was nearly tenfold when using the different limits, from US$500,000 (SP) to US$5 million (WHO). The higher cost in attention levels comes mainly

from the fact that more serious health impacts occur in higher concentration thresholds, which we considered as hospitalization costs (Table 3).

**Table 6.** Estimated costs for Emergency Room (ER) visits (exceedance of ozone standards) and ER visits plus hospitalization (exceedance of attention levels) for vulnerable and extremely vulnerable groups, in the areas of exceedance of the four standards used. Values are presented in US$.

| 8-h Ozone Average | 140 μgm$^{-3}$ | 100 μgm$^{-3}$ | 200 μgm$^{-3}$ | 160 μgm$^{-3}$ |
|---|---|---|---|---|
| **Standard** | **AQS SP** | **AQS WHO** | **ATT SP** | **ATT WHO** |
| **Vulnerable Population** | | | | |
| Elderly + children | 162,114 | 187,201 | 60,952 | 564,077 |
| Asthmatics | 1,408,302 | 1,760,351 | 524,097 | 4,877,639 |
| **Total Cost** | **1,570,416** | **1,947,552** | **585,049** | **5,441,716** |
| **Extremely Vulnerable Population** | | | | |
| Low income asthmatics with no external coating | 45,817 | 67,510 | 0 | 141,302 |
| Low income elderly + children | 36,204 | 51,575 | 0 | 112,235 |
| **Total Cost** | **82,021** | **119,085** | **0** | **253,537** |

AQS = air quality standard, ATT = attention levels, SP = State of São Paulo, WHO = World Health Organization.

We also performed a cost estimate for people in extreme vulnerability situation. In this case, we chose to use low-income asthmatics with no external coating and low-income age-sensitive groups. The WHO attention level was exceeded in low-income areas, whereas the current SP attention levels was only exceeded in areas with higher income (Figure 2b, also see Figure S2 for income). This shows a total cost of US$253,537 for WHO attention levels, which would not be considered at all by using the SP attention levels for the cost estimate. As vulnerabilities affect people in a synergic manner—a low-income child is under a higher risk of suffering respiratory symptoms than a higher-income child—we considered the extreme vulnerability groups as much more likely to suffer health effects, even compared the usual vulnerabilities group. Thus, the cost estimates for the extreme vulnerability groups should be considered as the minimum costs from ER visits and hospitalizations during the extreme ozone event, while the cost estimates from usual vulnerability groups should be considered as a potential average cost. These estimates are under considerable uncertainty, but we believe them to be very conservative because we considered the costs for one procedure for each person in the vulnerability groups. However, given the multiple times in which air quality limits were exceeded during this extreme event (Table 4), much worse health complications could be expected. Comparison with other studies of environmental and health valuation is difficult because, to the best of our knowledge, cost estimates from ER visits and hospitalizations caused by extreme air pollution events have not yet been performed for this area. Either way, healthcare planning for high air pollution levels can be severely miscalculated by using only the current SP state standards.

## 4. Discussion

Firstly, regional authorities must consider ozone a priority pollutant, as it still exceeds even the outdated SP state ozone standards. In addition, when considering the WHO limits, the number of exceedances increased much more in our simulation (from 37 to 55 exceedances). For the attention level exceedances, the number increased from five to 26 exceedances. This is probably due to the fact that the ozone attention level from the WHO (160 μgm$^{-3}$) is not much higher than what is the current standard in the SP state (140 μgm$^{-3}$). It becomes evident that, currently, the so-called "air quality standards" of SP state are close to very dangerous levels according to the international recommendations. Such pollutant levels should be treated as concentration limits to be avoided at all cost, due to their potential risks, not as a minimum acceptable standard of air quality. Concerning the difference in the size of the areas affected, caution must be taken when extrapolating model results, but our findings bring an insight into the severity of this problem. When applying stricter air quality standards, such as the WHO guidelines, the size of the area affected increases, and this should be carefully

considered due to the extreme population concentration in megacities. In these settings, an increase of a few km$^2$ corresponds to a potential threat to thousands or even millions of people (Tables 4 and 5).

The current air quality standards and attention levels must be updated in the SP state, particularly given the likelihood of increased extreme climatic events. A recent study showed that ozone exposure in Europe increased from 9% (2014) to 30% (2015), due to the strong positive temperature anomalies observed in that year over the European continent [36], implying that exposure to ozone can increase along with increasing air temperature in other regions characterized by high ozone levels, even more so in developing countries. However, another study points out that the ozone increase in urban areas is likely observed due to the decrease of the titration effect by the decrease in urban NOx concentrations [37]. This shows the nature of ozone pollution, influenced both by global and regional factors and calls for multiple-scale studies able to cope with this complexity.

If the WHO air pollution limits are to be attained, several improvements must be made. Policies involving the massive use of biofuels are constantly presented as more environmental-friendly options compared to fossil fuels, due to their renewability and lower emission factors of certain pollutants, such as CO. However, the use of biofuels may lead to higher VOC emissions. Many VOCs are ozone precursors, so, increased VOCs emission might result in an increase in ozone concentrations, according to their reactivity and the NOx/VOCs ratio [37–40]. Since ozone concentrations are still a major concern in the MASP and many other megacities in the world, the use of biofuels must be evaluated properly and implemented conjointly with policies that encourage the use of public transport systems, urban mobility improvements and electric vehicles, aimed to decrease vehicle activity and total emissions. In this context, public policies must be integrated in all levels of public management to provide the best conditions for mitigating emissions, concentrations and impacts of air pollution, according to their responsibilities—federal (fuel improvements, etc.), state-level (establishment of truly protective air quality limits, intercity transport) and municipal (proper healthcare management, urban transport). Policymakers must account for such events and guarantee proper investments in public health capable to deal with their consequences, such as medical procedure costs from respiratory conditions. Planning for cost must be a priority and using outdated air quality standards will only hurt long-term planning, therefore, the establishment of more rigid air quality standards is crucial.

Regarding the low-income population, whenever extreme events are forecasted or happen unexpectedly, allocating extra health agents in public hospitals and healthcare facilities near the exceedance areas will certainly help to deal with the higher number of patients seeking public health services. Public policies aimed to improve poor housing conditions, providing external coating, better isolation from outdoor air and overall cleaner indoor conditions, have the potential to decrease pollution and environmental exposure. For the asthmatics group, the results from this study provide an idea of how much cost can be avoided during extreme events if asthma is better controlled with medications, and fewer ER visits and hospitalizations are required.

## 5. Conclusions

In this study, we used the WRF-Chem model to simulate an extreme ozone event triggered by unusually sunny and warm conditions in the metropolitan area of Sao Paulo in 2014. We then demonstrated how different air pollution limits correspond to different scenarios of exposure for vulnerable groups and their impacts on public health cost planning during the episode. Such events are possibly more likely to occur in areas with an abundance of ozone precursors due to climate change because ozone is strongly associated with atmospheric conditions. The frequent local wind system of the sea breeze has the potential to transport air pollution to areas located downwind characterized by low income, inhabited by vulnerable population groups. It is clear from these results that policymakers should prioritize these areas for environmental management and public health in extreme ozone pollution events.

Global climate trends produce different local impacts according to the social and environmental characteristics of each area and it is paramount to understand such interactions in a fast-changing world.

In this study, it became clear that the current air quality standards in the state of São Paulo severely misrepresent potential population exposure to harmful air pollution levels. When the recommended limits from the WHO were applied to the results of our simulation, the area affected by violations of the Air Quality Standards increased considerably (from 2931 to 4138 km$^2$), along with the estimated number of people exposed. More than two million people in low-income conditions live in areas affected by ozone concentrations above the WHO attention levels, a risk not detected by applying the current SP limits. The exposure of the low-income population to environmental hazards, such as extreme atmospheric and air pollution events, has complex origins, generates systemic problems and must be confronted with equally integrated approaches. The combination of policies which ignore the WHO ozone attention levels with low-income population exposed to those limits brings about an alarming situation. The greatest difference in public health cost estimates for this event was from US$585,049 (considering SP state attention levels) to US$5,441,716 (considering WHO attention levels), nearly a tenfold increase, which suggests a mismatch in proper public health cost planning. This calls for an urgent update of the SP state air quality standards, representative of the full vulnerability spectrum of the population across a megacity in a developing nation.

Concerning future works, other pollutants also reach dangerous levels due to atmospheric stagnation in such extreme events, not to mention health effects from the extreme atmospheric conditions per se. This calls for integrative approaches, considering the vulnerability and health effects of air pollution and extreme climatic conditions conjointly. Other countries or regions not following the WHO air quality guidelines are encouraged to perform similar analyses in order to understand the importance of updating air quality guidelines for proper impact assessment, particularly considering the Global South.

**Supplementary Materials:** The following are available online at http://www.mdpi.com/2071-1050/11/13/3725/s1, Figure S1: Population density in the MASP weighting areas, Figure S2: Low-income weighting areas in the MASP (monthly family income lower than USD 500), Figure S3: (a) Location of the metropolitan area of São Paulo, its main road network and model domain (excluding the sparsely populated areas to the far east); (b) simplified example of emission file in the WRF/Chem model domain (green in Figure S3a): emission of nitrogen monoxide (NO) at 19 h, Table S1: Parametrizations used in WRF/Chem.

**Author Contributions:** Conceptualization, J.B.C., M.E.S.S., S.A.I.-E. and R.Y.Y.; data curation, M.E.S.S. and S.A.I.-E.; formal analysis, J.B.C. and W.C.-M.; funding acquisition, J.B.C. and M.E.S.S.; investigation, J.B.C. and R.Y.Y.; methodology, J.B.C., M.E.S.S., W.C.-M., F.N.D.R., S.A.I.-E. and R.Y.Y.; project administration, J.B.C. and M.E.S.S.; resources, M.E.S.S.; software, J.B.C., M.E.S.S., W.C.-M., S.A.I.-E. and R.Y.Y.; supervision, M.E.S.S. and R.Y.Y.; validation, J.B.C., M.E.S.S. and F.N.D.R.; visualization, J.B.C. and W.C.-M.; writing—original draft, J.B.C.; writing—review and editing, J.B.C., M.E.S.S., F.N.D.R., S.A.I.-E. and R.Y.Y.

**Funding:** This research was funded by FAPESP (Research Funding Agency of the São Paulo State), grant number 2012/12216-5.

**Conflicts of Interest:** The authors declare no conflict of interest. The funders had no role in the design of the study; in the collection, analyses, or interpretation of data; in the writing of the manuscript, or in the decision to publish the results.

## Appendix A

Detailed WRF/Chem simulation information according to the scheme of Andrade et al., 2015 [28] and validated in Chiquetto et al., 2018 [6].

- Four vehicle categories: light, trucks, buses and motorbikes:
- Km driven per day for each category
- Emission factors—gasoline (with 22% ethanol), ethanol, flex-fuel and diesel
- Proportion of vehicles per type
- Chemical mechanism RADM2
- Photolysis Madronich [41]
- Reactions Stockwell [42]
- Ethanol as an explicit chemical species

- Period: 28/01–02/02/2014 (anomalies of temperature (+3 °C) and OLR (+60 Wm$^{-2}$))
- Spatial resolution: land use/surface 30 s (approximately 800 m)/emissions 1 km
- Spatial resolution (timestep): 6 s

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
