# Peer review of "Air Quality Standards and Extreme Ozone Events in the São Paulo Megacity"

_sustainability, doi:10.3390/su11133725_

Reviewer 1 Report

The current version needs to be strongly/deeply revised before publication. The current version is similar to a technical report.

The introduction and section M&M need to be deeply shortened (5 pages) and the section "results" (4 pages) compared to the section "Discussion" (1 page). This last section is scientifically weak with no references.

The list of keywords should be revised and limited. In introduction, a state-of-the-art in US and Europe should be added as well as used standards for human health protection.

The introduction is too long, plenty of well-known statements. The sub-titles must be removed. For instance the sentence L55-60 can be summarized as 1 line "Several studies ... vulnerable groups [8-10]" => Lower income population, elderly and children  are more vulnerable to the effects of air pollution, similarly to São Paulo".

L74-78 - The ozone concentrations are rising in cities (Sicard et al.) due to lower ozone titration by NO. L97-138 => to be shortened !

The section m&M is boring.

Author Response

Response to Reviewer 1 Comments

 Point 1: The current version needs to be strongly/deeply revised before publication. The current version is similar to a technical report. The introduction and section M&M need to be deeply shortened (5 pages) and the section "results" (4 pages) compared to the section "Discussion" (1 page). This last section is scientifically weak with no references.

Response 1: We appreciate the reviewer’s feedback. The comments are answered in red in the text below.

The text has been deeply revised. We have rewritten the whole introduction and methodology sections. Figure 1 was moved to the supplementary materials. We also made substantial changes to the other sections, aiming to a more concise and comprehensible text, according to the number of pages suggested by the reviewer. We highlighted the new text in red. Concerning the discussion section, we discuss public policies recommendations from our findings, aiming to better prepare the authorities for environmental and health costs planning. We also discuss the influence of climate change and NOx in ozone concentrations, and the issues concerning the use of biofuels with increased VOC emissions. Five references are cited.

Point 2: The list of keywords should be revised and limited. In introduction, a state-of-the-art in US and Europe should be added as well as used standards for human health protection.

Response 2: The journal requires authors to include three to ten keywords, so we included ten keywords with topics directly related to the study. We can accept suggestions if the reviewer feels they are not appropriate.

Table 1 has been inserted in the introduction section with the air quality limits from the São Paulo state and the World Health Organization used in the study.

An analysis of the state-of-the-art has been done about the studies in this topic (lines 67-90) including studies in the US and Europe, Brazil, Australia and South Korea. The references used are listed below:

Amann, M. Health risks of ozone from long-range transboundary air pollution; World Health Organization Regional Office for Europe: Copenhagen, Denmark, 2008.

Knowlton K.; Rotkin-Ellman M.; Geballe L.; Max W.; Solomon G.M. Six climate change–related events in the United States accounted for about $14 billion in lost lives and health costs. Health Aff. 2011, 30(11), 2167-2176. https://doi.org/10.1377/hlthaff.2011.0229

 Costa, E.; Caetano, R.; Werneck, G.L.; Bregman, M.; Araújo, D.V.; Rufino, R. Estimated cost of asthma in outpatient treatment: a real-world study. Rev. Saúde Pública 2018, 52, 27 https://doi.org/10.11606/S1518-8787.2018052000153

 Abe K.; Miraglia S. Health impact assessment of air pollution in São Paulo, Brazil. Int. J. Environ. Res. Public Health 2016, 13(7), 694. https://doi.org/10.3390/ijerph13070694

 Kolbe A.; Gilchrist K.L. An extreme bushfire smoke pollution event: health impacts and public health challenges. N. S. W. Public Health Bull. 2009, 20(2), 19-23. https://doi.org/10.1071/NB08061

 Bae H.J.; Kang J.E.; Lim Y.R. Assessing the Health Vulnerability Caused by Climate and Air Pollution in Korea Using the Fuzzy TOPSIS. Sustainability. 2019, 11(10), 2894.

 Brandt J.; Silver J.D.; Christensen J.H.; Andersen M.S.; Bønløkke J.H.; Sigsgaard T.; Geels C.; Gross A.; Hansen A.B.; Hansen K.M.; Hedegaard G.B. Contribution from the ten major emission sectors in Europe and Denmark to the health-cost externalities of air pollution using the EVA model system–an integrated modelling approach. Atmos. Chem. Phys. 2013, 13(15), 7725-7746. https://doi.org/10.5194/acp-13-7725-2013

 Please let us know if this list is not appropriate and why, and if there are any other interesting suggestions for this section.

 Point 3: The introduction is too long, plenty of well-known statements. The sub-titles must be removed. For instance the sentence L55-60 can be summarized as 1 line "Several studies ... vulnerable groups [8-10]" => Lower income population, elderly and children  are more vulnerable to the effects of air pollution, similarly to São Paulo".

 Response 3: We appreciate your instructions and the introduction has been adjusted. We rewrote the mentioned sentence as follows: “Several studies worldwide have investigated air pollution exposure according to socioeconomic conditions, showing that lower income population, elderly and children are more vulnerable to the effects of air pollution [9-12], similarly to São Paulo [13-15].”

Point 4: The ozone concentrations are rising in cities (Sicard et al.) due to lower ozone titration by NO. L97-138 => to be shortened !

 Response 4: We thank the reviewer for the suggestions and have added the requested reference in the context of our study, mentioning the effects of NOx titration on ozone. However, we would like to inform that the period of the recommended study (2000-2010) is different from the period analysed in the reference we had cited, concerning the effects of climate change over ozone during more recent years (2014-2015), which match with the period of our study. We have adjusted this text and relocated it to the discussion section, according to our response to the comments of another reviewer.

 Point 5: The section m&M is boring.

 Response 5: The text in this section has been adjusted.

 We are very thankful for all useful comments, adjustments and improvements from the reviewer, and we are open to any further inquiry they should have. Please find the revised version of the MS and we hope that it is suitable for publication.

Reviewer 2 Report

Dear Authors,

My comments are uploaded, see the attachment. 

Reviewer

Author Response

Response to Reviewer 2 Comments

 The authors applied a model simulation to an extreme ozone event in the São Paulo Megacity during an extreme high air temperature and solar radiation event. Then they evaluated the results using different air quality thresholds, as well as local socioeconomic vulnerability data and a cost analysis for the public health system from the extreme episode.

This is an interesting manuscript. Air pollution is an ever-increasing problem and it is important to deal with any aspect of it. Hence, these kinds of studies are of great importance.

Especially, I appreciate that not a primary (diseases) but a secondary consequence of the problem (vulnerability and costs) – a less thoroughly analysed area – is examined in the paper.

This is a thorough analysis on the area.

We appreciate the reviewer’s proper corrections. We are very thankful for the positive feedback on our work. The comments are answered in red in the text below, and the corresponding changes are also highlighted in red in the new version of the MS text. The text has been deeply revised as per the comments of the other reviewers.

 My remarks are as follows.

 Comments:

 Point 1: row 141: correctly “irradiance”, instead of “insolation”. 

 Response 1: Corrected.

 Point 2: correctly “air pressure”, instead of “pressure”;.

 Response 2: Corrected.

 Point 3: correctly “Subtropical Atlantic High Pressure System”, instead of “Subtropical Atlantic High”;

 Response 3: Corrected.

 Point 4: row 172: correctly: “parameterizations”, instead of “parametrizations”;

 Response 4: Corrected.

 Point 5: page 5, in the explanation of eqs. 1-4: correctly “TH”, instead of “TH”, and correctly “TH-n” instead of “TH-n”;

 Response 5: Corrected.

 Point 6: correctly, throughout the manuscript, “criterion” instead of “criterium”;

 Response 6: Corrected.

 Point 7: correctly: “population density maps” instead of “populational density maps”;

 Response 7: Corrected.

 Point 8: rows 339-341: It is not clear for me what are the two clusters between which you calculate Pearson linear correlation coefficient. Please explain it in more detail! In addition, it is not apparent from the text whether this correlation coefficient is significant or not at the 5% probability level. Besides the selected probability level, significance depends on the number of element pairs you compare. Even though the correlation coefficient seems to be high, if the number of element pairs compared is little, the correlation coefficient may not be significant at the selected probability level.

 Response 8: We thank the reviewer for the comment, and in fact, we did not approach this correctly. The correlation was calculated using the weighting areas for the whole metropolitan region, shown in the various maps in the paper, we performed a spatial correlation between the variables using the data from the weighting areas. The 0.8 correlation coefficient between “above-average number of children” and “houses with no external coating” is statistically significant at the level of 1%. However, we did not present any other results from the correlations, among other socioeconomic/vulnerability variables and with air pollution in the different limits, so this information is not truly integrated into the paper’s main results and the overall methodology/framework. That being said, we decided to suppress this information in this paper. Also, the other reviewers required that the whole text of the MS must be shortened, and unfortunately, there is no more room in the paper to properly explore the spatial correlations and all the required adjustments it would need. We hope the reviewer understands these necessary adjustments in the MS text.

 Point 9: I think that may apply a multiple statistical analysis for detecting hidden relationships among the explanatory variables you use and other variables you may consider as target variables

 Response 9: We appreciate the reviewer’s considerations and strategic views on our findings. It is not clear if this suggestion is meant to be for this study or a future study. But, considering that there are already many different types of results in this paper (air quality violations, area size, number of people affected and public health costs from two different environmental legislations) and the demand from the other reviewers to shorten the text considerably, we conclude that there is not enough room in this manuscript to include a new set of results which would demand the appropriate adjustments in the whole MS text. In fact, we are already working in another manuscript with the results of the spatial correlations between air pollution and environmental exposure variables in a different context and we plan to submit it soon.

 Point 10: row 368: Table 4: It is not clear for me, what are the criteria of extreme vulnerability?

 Response 10: We added a short section explaining the extreme vulnerability groups right before the insertion of table 5 (previously table 4), lines 264 to 267, as follows:

We considered the vulnerable population groups as explained in table 2; two additional groups were considered as extremely vulnerable, where one or more vulnerability criteria stacked and applied to the same population (in this case, low-income asthmatics with no external coating and low-income age-sensitive groups)”.

 Point 11: Table 4: column 1 should be left closed;

 Response 11: Corrected, please check.

 Point 12: Table 4, first row, columns 2. 3. 4 and 5: the columns should be wider so that the exponent “3” does not get into the next row.

 Response 12: Corrected.

 Point 13: Table 4, first row, columns 2. 3. 4 and 5: correctly “μgm-3”, instead of “μg・m-3”; namely, dot should not be used between “μg” and “m-3”. Instead, can choose: you may use nothing (“μgm-3”) or you use multiplication sign between “μg” and “m-3”, namely (“μg・m-3”).

 Response 13: Corrected.

 Point 14: row 405: This is Table 5. Please correct it.

Response 14: Corrected.

 Point 15: Table 5: All my comments mentioned concerning Table 4 is valid to Table 5, too. Please improve Table 5 accordingly.

 Response 15: Corrected, please check.

 Point 16: row 442: correctly “to the best of our knowledge”, instead of “to the best of your knowledge”;

 Response 16: Corrected.

 Point 17: row 461: correctly “corresponds”, instead of “correspond”;

 Response 17: Corrected.

Point 18: correctly “due to their” instead of “due their”;

 Response 18: Corrected.

 Point 19: Figure S1: The title of the figure within the frame of the figure is duplicate. Please delete it.

 Response 19: Corrected.

 Point 20: Figure S1, legends: correctly “inhabitants / km2” instead of “people / km2”; it is stressed again that “2” in “km2” is an exponent!

 Response 20: Corrected.

 Point 21: Figure S2: see my comment, regarding the title, to Figure S1.

 Response 21: Corrected.

 Point 22: Table S1: the first column titled “Physics Options” should be left closed;

 Response 22: Corrected, please check.

We are very thankful for all useful comments, adjustments and improvements from the reviewer, and we are open to any further inquiry they should have. Please find the revised version of the MS and we hope that it is suitable for publication.

Reviewer 3 Report

This is an interesting research paper concerning air pollution in Sao Paulo Region Brazil.

Major points:

The paper contains too much information that are not comprehensible for the reader.

The mixture of ozone air pollution, social and environmental vulnerability as well a climate change is much too difficult to understand. The paper needs therefore more clear structure.

It needs kind of an unbundling. The aspect of Climate change is anyway too superficial and might be abandoned.

The conclusion part of the paper is very generic written. What is the aim of the authors? Who should do what excately?

Minor points:

The title is too long and too complicated.

The language including all abbreviations must be carefully checked

Author Response

Response to Reviewer 3 Comments

This is an interesting research paper concerning air pollution in Sao Paulo Region Brazil.

 We appreciate the reviewer’s feedback and the interest in our work. The comments are answered in red in the text below, and the corresponding changes are also highlighted in red in the new version of the MS text. Figure 1 was moved to the supplementary materials. The text has been deeply revised as per the comments of the other reviewers.

 Point 1: The paper contains too much information that are not comprehensible for the reader.

Response 1: We have rewritten the whole introduction and methodology sections, aiming to a more concise and comprehensible text. For example, we changed the acronym AQS (Air Quality Standard) to simply ‘ozone standard’ in all the text (except in the tables where space is needed). We have also made changes to the other sections and highlighted the new text in red. If any there any other issues left, we politely ask the reviewer to point out what is not comprehensible in the text.

 Point 2: The mixture of ozone air pollution, social and environmental vulnerability as well a climate change is much too difficult to understand. The paper needs therefore more clear structure. It needs kind of an unbundling. The aspect of Climate change is anyway too superficial and might be abandoned.

 Response 2: We appreciate the suggestions. The text has been reviewed. The climate change aspect has been adjusted and relocated to the discussion section. According to the scientific literature and the references provided, ozone is strongly associated with atmospheric conditions, so we feel that this aspect cannot be abandoned in this manuscript, which is a simulation of an extreme event.

 Point 3: The conclusion part of the paper is very generic written. What is the aim of the authors? Who should do what excately?

 Response 3: The objective of this paper is described in lines 91-97:

 "The objective of this work is to show the difference in the number of exceedances of the air quality standards and attention levels, in the area size, the difference in the number of people affected, including those in vulnerability situations and a cost analysis for the public health system, during an extreme ozone event in the MASP, from using outdated air quality standard. Results are compared using the Air Quality Standards and Attention Levels from the WHO and the SP state. We discuss the consequences of the current lenient air quality limits of the SP State for environmental and health management and planning”.

 The conclusion section of the paper has been revised. We include below some parts of the text concerning the reviewer’s comments:

 -Lines 389-395: “In this study, it became clear that the current Air Quality Standards in the state of São Paulo severely misrepresents potential population exposure to harmful air pollution levels. When the recommended limits from the WHO were applied to the results of our simulation, the area affected by violations of the Air Quality Standards increased considerably (from 2931 to 4138 km2), along with the total number of exposed people. More than two million people in low-income conditions live in areas affected by ozone concentrations above the WHO Attention Levels, which were completely overlooked by the current SP limits”.

 -Lines 402-403: “This calls for an urgent update of the SP state Air Quality Standards, representative of the full vulnerability spectrum of the population across a megacity in a developing nation”.

The more specific discussions of public policies derived from the study were too lengthy for the conclusion section, so we included them in the discussion section, lines 343-377:

 “The current Air Quality Standards and Attention Levels must be updated in the SP state, particularly given the likelihood of increased extreme climatic events. A recent study showed that ozone exposure in Europe increased from 9% (2014) to 30% (2015), due to the strong positive temperature anomalies observed in that year over the European continent [35], implying that exposure to ozone can increase along with increasing air temperature in other regions characterized by high ozone levels, even more so in developing countries. However, another study points out that the ozone increase in urban areas is likely observed due to the decrease of the titration effect by the decrease in urban NOx concentrations [36]. This shows the complexity of ozone pollution, influenced both by global and regional factors.

 If the WHO air pollution limits are to be attained, several improvements must be made. Policies involving the massive use of biofuels are constantly presented as more environmental-friendly options compared to fossil fuels, due to their renewability and lower emission factors of certain pollutants, such as CO. However, the use of biofuels may lead to higher VOC emissions. Many VOCs are ozone precursors, so, increased VOCs emission might result in an increase in ozone concentrations, according to their reactivity and the NOx/VOCs ratio [37-39]. Since ozone concentrations are still a major concern in the MASP and many other megacities in the world, the use of biofuels must be evaluated properly and implemented conjointly with policies that encourage the use of public transport systems, urban mobility improvements and electric vehicles, aimed to decrease vehicle activity and total emissions. In this context, public policies must be integrated in all levels of public management to provide the best conditions for mitigating emissions, concentrations and impacts of air pollution, according to their responsibilities – federal (fuel improvements, etc.), state-level (establishment of truly protective air quality limits, intercity transport) and municipal (proper healthcare management, urban transport). Policymakers must account for such events and guarantee proper investments in public health capable to deal with their consequences, such as medical procedure costs from respiratory conditions. Planning for cost must be a priority and using outdated air quality standards will only hurt long-term planning, therefore, the establishment of more strict air quality standards is crucial.

Regarding the low-income population, whenever extreme events are forecasted or happen unexpectedly, allocating extra health agents in public hospitals and healthcare facilities will certainly help dealing with the higher number of patients seeking public health services. Public policies aimed to improve poor housing conditions, providing external coating, better isolation from outdoor air and overall cleaner indoor conditions, have the potential to decrease pollution and environmental exposure. For the asthmatics group, the results from this study provide an idea of how much cost can be avoided during extreme events if asthma is better controlled with medications, and less ER visits and hospitalizations are required”.

 Point 4: The title is too long and too complicated.

 Response 4: We revised the title and changed to “Air Quality Standards and Extreme Ozone Events in the São Paulo Megacity”.

 Point 5: The language including all abbreviations must be carefully checked

 Response 5: We have reviewed the language when adjusting the text to its new, shorter version. The abbreviations have been checked (ex; ‘AQS’ has been changed to ‘ozone standard’ and µg.m-3 has been changed to µgm-3, according to the suggestions of another reviewer).

 We are very thankful for all useful comments, adjustments and improvements from the reviewer, and we are open to any further inquiry they should have. Please find the revised version of the MS and we hope that it is suitable for publication.

Round  2

Reviewer 1 Report

all requested changes are included.

Reviewer 2 Report

Dear Authors,

Iaccept the response of the authors to my comments and also the revised manuscript. 

Reviewer

Reviewer 3 Report

The authors revised the paper according to the instructions of the reviewer and the paper is now ready for publication.